# Preclinical PET Imaging of Tumor Cell Death following Therapy Using Gallium-68-Labeled C2Am

**DOI:** 10.3390/cancers15051564

**Published:** 2023-03-02

**Authors:** Flaviu Bulat, Friederike Hesse, Bala Attili, Chandra Solanki, Iosif A. Mendichovszky, Franklin Aigbirhio, Finian J. Leeper, Kevin M. Brindle, André A. Neves

**Affiliations:** 1Cancer Research UK Cambridge Institute, University of Cambridge, Cambridge CB2 1TN, UK; 2Department of Chemistry, University of Cambridge, Cambridge CB2 1EW, UK; 3Addenbrooke’s Hospital Radiopharmacy, Cambridge University Hospitals NHS Foundation Trust, Cambridge CB2 0QQ, UK; 4Department of Nuclear Medicine, Cambridge University Hospitals NHS Foundation Trust, Cambridge CB2 0QQ, UK; 5Department of Radiology, University of Cambridge, Cambridge CB2 1EW, UK; 6Wolfson Brain Imaging Centre, University of Cambridge, Cambridge CB2 0QQ, UK; 7Department of Biochemistry, University of Cambridge, Cambridge CB2 1GA, UK

**Keywords:** cell death, apoptosis, tumor, imaging, C2Am, TRAIL-R2, positron emission tomography (PET), gallium-68

## Abstract

**Simple Summary:**

There is an unmet clinical need for imaging agents capable of detecting the presence, extent, and distribution of tumor cell death following treatment. We describe here a gallium-68-labeled derivative of the C2A domain of Synaptotagmin-I (^68^Ga-C2Am), which binds to the phosphatidylserine exposed by dying cells, for imaging tumor cell death in vivo using positron emission tomography (PET). Since PET is a highly sensitive tomographic imaging technique that is widely used in clinical oncology and gallium-68 has emerged as a cost-effective radiotracer that can be eluted on site from a benchtop generator, ^68^Ga-C2Am could find clinical application for the rapid assessment of tumor responses to treatment.

**Abstract:**

There is an unmet clinical need for imaging agents capable of detecting early evidence of tumor cell death, since the timing, extent, and distribution of cell death in tumors following treatment can give an indication of treatment outcome. We describe here ^68^Ga-labeled C2Am, which is a phosphatidylserine-binding protein, for imaging tumor cell death in vivo using positron emission tomography (PET). A one-pot synthesis of ^68^Ga-C2Am (20 min, 25 °C, >95% radiochemical purity) has been developed, using a NODAGA-maleimide chelator. The binding of ^68^Ga-C2Am to apoptotic and necrotic tumor cells was assessed in vitro using human breast and colorectal cancer cell lines, and in vivo, using dynamic PET measurements in mice implanted subcutaneously with the colorectal tumor cells and treated with a TRAIL-R2 agonist. ^68^Ga-C2Am showed predominantly renal clearance and low retention in the liver, spleen, small intestine, and bone and generated a tumor-to-muscle (T/m) ratio of 2.3 ± 0.4, at 2 h post probe administration and at 24 h following treatment. ^68^Ga-C2Am has the potential to be used in the clinic as a PET tracer for assessing early treatment response in tumors.

## 1. Introduction

Imaging allows serial assessment of tumor responses to treatment in the clinic, where early detection of response can allow rapid selection of the most effective therapy (personalized treatment), avoiding the deleterious side effects of ineffective therapy and reducing healthcare costs [1]. Currently, tumor treatment response is evaluated using morphological criteria, as described by the Response Evaluation Criteria in Solid Tumors 1.1 (RECIST) [2]. However, morphology-based assessment is slow, and it can take several weeks to reach a clinical conclusion. Differentiating between pseudo-progression and true progression is also a challenge. The difficulties in obtaining an early indication of treatment response from changes in tumor size have been addressed by using functional imaging techniques, such as positron emission tomography (PET), which is a widely available clinical imaging modality that is highly sensitive, quantitative, and provides tomographic data. Changes in tumor metabolism post-treatment often precede decreases in tumor size [1]. These can be detected, for example, using PET measurements of uptake of the glucose analog [^18^F]fluoro-D-deoxyglucose (^18^F-FDG) and guidelines for its use in assessing treatment response have been published (PET Response Evaluation Criteria in Solid Tumors (PERCIST) [3]). However, the technique has some limitations including high uptake in normal brain tissue, which can hinder response assessment in brain tumors, and fast renal excretion and accumulation in the bladder, which can impair response assessment in tumors in the adjacent prostate. Inflammatory responses post-treatment can also lead to increased uptake and delaying imaging by several weeks may be required to allow the inflammation to clear [4].

Tumor cell death post-treatment has been identified as a good prognostic indicator for treatment outcome [1,5]. However, in the clinic, detection of cell death is often based on histological assessment of biopsy samples, which is invasive, provides only a narrow spatio-temporal assessment of the lesion’s status, and ignores tumor heterogeneity. Imaging can provide a non-invasive and longitudinal assessment of cell death within the entire tumor volume. Several PET imaging agents have been developed that target cell death-related events [6,7,8], some of which have entered early stage clinical testing, albeit with limited success. During apoptosis, caspase 3 is activated by proteolytic cleavage to give cleaved caspase 3 (CC3), which has been targeted using small molecules such as the fluorine-18 labeled isatin derivative [^18^F]ICMT-11 [9]. Since caspase activation is a transient phenomenon imaging must be performed within a narrow timeframe, which can be highly variable and dependent on tumor type. Thus, imaging protocols that use this agent will require individual optimization. In clinical studies with [^18^F]ICMT-11, a lack of tracer accumulation in breast tumors was attributed to low levels of apoptosis post-treatment, with only 1% of cells becoming apoptotic. However, the voxel-wise analysis showed increases in [^18^F]ICMT-11 retention in regions of some tumors. 18F-ML-10, an agent that detects the alterations in membrane permeability that accompany cell death, failed to detect response to chemotherapy in a pre-clinical model of breast cancer [10]. A recently described agent based on a ^68^Ga-labeled tripeptide trivalent arsenical (4-(N-(S-glutathionylacetyl)amino)phenylarsonous acid) which enters dying cells and cross-links cysteines 597 and 598 of heat shock protein 90, thus trapping the agent in the dying cell [11], has been used recently in a first in man study [12]. However, there are no data as yet on the effectiveness of this PET-detectable agent for imaging cell death in the clinic.

The phospholipids PS and phosphatidylethanolamine (PE) which are normally confined to the inner leaflet of the plasma membrane bilayer are exposed on the outer leaflet during apoptosis [13], or during necrosis by the enhanced accessibility that occurs following plasma membrane disruption. These represent abundant and easily accessible imaging targets that could potentially provide good image contrast. Exposed PE has been detected using duramycin, a 2 kDa tetracyclic peptide that has been labeled with the γ-emitter ^99m^Tc [14] for detection using SPECT, and with ^18^F [15] and ^68^Ga [16] for detection using PET. Exposed PS has been detected using annexin V, a 36 kDa protein that binds PS with sub-nanomolar affinity [17] and has been the most widely used imaging agent for detecting cell death, with a plethora of derivatives designed for fluorescence, MRI, PET, and SPECT detection in preclinical and clinical studies [18]. A SPECT (99mTc-labeled) derivative of annexin V, which binds PS, entered Phase 1 and 2 clinical trials two decades ago, however, it was discontinued due to poor pharmacokinetics and non-specific abdominal and kidney retention [19]. In preclinical studies, two PET-detectable ^68^Ga-labeled derivatives of annexin V were produced by site-specific labeling at introduced cysteine residues. The derivatives showed rapid, predominantly renal clearance, with short blood half-lives of ~6 min, however, kidney retention was high [20]. Annexin V labeled on amino groups with N-succinimidyl 4-[^18^F]fluorobenzoate showed lower uptake in the liver, spleen, and kidney than the ^99m^Tc-labeled protein [21] and more recently annexins labeled with ^18^F at specific cysteine residues introduced by site-specific mutagenesis have been described [22,23,24].

The C2A domain of the vesicle-binding protein Synaptotagmin is a small 16 kDa protein that, like annexin V, binds PS with nanomolar affinity [25]. The protein was used initially as a fusion protein with glutathione S-transferase (GST) and labeled on amino groups with super-paramagnetic iron oxide nanoparticles (SPIO) [26] and a Gd^3+^-chelate for MRI detection [27] and with ^99m^Tc for detection using SPECT [28]. Subsequently, the isolated C2A domain was used in which a single cysteine residue was introduced for site-specific labeling [25] and the protein (designated C2Am) has been labeled with ^99m^Tc and ^111^In for SPECT detection [29], and with a fluorescent dye for detection using photoacoustic imaging [30]. A fluorescently labeled C2Am derivative was compared with a similarly labeled annexin V and although it showed a lower affinity for binding to PS (Kd = 71 nM) it showed similar labeling of necrotic and apoptotic cells but with lower binding to viable cells and therefore better specificity for detecting cell death [25].

We have shown recently that C2Am when labeled with the positron-emitting isotope ^18^F, can be used to detect tumor cell death using PET with high sensitivity and specificity at 24 h post-treatment and within one hour of systemic administration [31]. However, labeling of C2Am with the short-lived ^18^F radioisotope (t_1/2_ = 109.8 min) requires a nearby cyclotron facility. This is a large and expensive piece of equipment that has high operational and maintenance costs [32]. Gallium-68 (t_1/2_ = 67.8 min) is a cost-effective alternative to ^18^F since it can be obtained from a benchtop generator [33]. Several ^68^Ga-labeled imaging agents have been approved recently for the clinical monitoring of advanced oncological disease, including ^68^Ga-PSMA [34], ^68^Ga-DOTATE [35], and ^68^Ga-DOTANOC [36]. We describe here ^68^Ga-C2Am, a gallium-68-labeled C2Am derivative for in vivo PET imaging of tumor cell death post-treatment.

## 2. Materials and Methods

### 2.1. Production of ^68^Ga-C2Am

The chelator NODAGA-maleimide (Chematech, Dijon, France) was conjugated to the single cysteine in HBS buffer (20 mM HEPES, 150 mM NaCl, pH 7.4), 8-fold molar excess chelator: C2Am, 30 min, pH 7.4, 25 °C and then manually loaded with [^68^Ga]GaCl_3_ eluted from a generator in a one-pot synthesis (5 min, pH 6.0, 25 °C, >95% radiochemical purity and yield, specific activity (SA) 0.16 MBq/μg of protein; Figure 1, Appendix A). Radiosynthesis was carried out manually using a lead castle for shielding. NaOH (0.1 M) was used to adjust the final pH to 7.0–7.5. Activity yield (AY) at the end-of-synthesis (EOS) was >95% due to the short incubation time. Activities of the final product ranged between 28–34 MBq and were dependent on the activity of generator eluate.

Quality control consisted of ultra-performance liquid chromatography using a Superdex 75 Increase 5/150 gel filtration column (Cytiva, Emeryville, CA, USA), which is designed for separating low molecular weight proteins (3 kDa to 70 kDa). Small molecules, including [^68^Ga]GaCl_3_, have long retention times (>10 min).

### 2.2. Cell Culture

Media components were obtained from Sigma-Aldrich unless stated otherwise. COLO205 (CCL-222^®^) and MDA-MB-231 (HTB-26^®^) were purchased from ATCC (Gaithersburg, MD, USA) and authenticated using short-tandem repeat genetic profiling yielding a 100% match to the ATCC database. Cells were used within ten passages from the original stocks and were tested regularly for mycoplasma. COLO205 and MDA-MB-231 cells were cultured in RPMI-1640 and DMEM medium, respectively, supplemented with 10% fetal bovine serum (Life Technologies, Carlsbad, CA, USA) and 2 mmol/L L-glutamine at 37 °C and 5% CO2.

### 2.3. Dead Cell Labeling with ^68^Ga-C2Am

Drug vehicle-treated and MEDI3039-treated (10 pM, 24 h) COLO205 or MDA-MB-231 cells were harvested, viability assessed and counted using an automated trypan blue stain analyzer (Vi-CELL^®^, Beckman Coulter Inc., Brea, CA, USA). Cells were resuspended in HBS buffer and incubated with 68Ga-C2Am (1–2 μM, 2.5–4 MBq, molar activity (A_m_) = ~2.7 GBq/μmol) at 37 °C for 20 min. Cell pellets (1 million cells before treatment, 400 g, 5 min, 4 °C) were washed three times with HBS buffer and radioactivity was counted using a gamma counter (AMG Hidex; 1 min) configured to monitor gallium-68 emission (511 keV). The percentage of retained activity was standardized to cell number for MDA-MB-231 and against total cell membrane area for COLO205, due to the formation of small apoptotic bodies following MEDI3039 treatment [31]. COLO205 apoptotic bodies have a smaller diameter (*D_apo_* = 9.2 μm, measured by Vi-CELL) than the corresponding intact cells (*D_int_* = 15.5 μm). The total outer cell membrane area (*A_t_ = N A_i_*) was, therefore, calculated from the total cell (or apoptotic body) number (*N*) multiplied by the individual cell (or apoptotic body) surface area (*A_i_*). Assuming a spherical geometry, *A_i_* = π *D_i_*
^2^, where *D_i_* is the diameter of an intact cell (*D_int_*) or apoptotic body of COLO205 cells (*D_apo_*).

### 2.4. Animal Studies

Mouse experiments were designed with reference to the UK Co-ordinating Committee on Cancer Research guidelines for the welfare of animals in experimental neoplasia [37] and conducted under a Project License issued under the Animals Scientific Procedures Act of 1986. Protocols were approved by an Animal Welfare and Ethical Review Body. COLO205 cells (5 million) were resuspended in phosphate-buffered saline (PBS) (0.1 mL, 137 mM NaCl, 2.7 mM KCl, 8 mM Na_2_HPO_4_, and 2 mM KH_2_PO_4_) and implanted subcutaneously in the upper back of 10–12-week-old female BALB/c Nu/Nu mice (Charles River). Tumors were imaged when they reached ~1 cm^3^.

Mice were anesthetized in an induction chamber prior to PET/CT scanning using isoflurane (3–5% *v*/*v*, Isofane^®^, Henry Schein, Melville, New York, USA) in a mixture of air and oxygen (1:1, at 1 L/min). Tail veins were then cannulated (BD Microlance 3, 30G needle) and flushed with saline (0.1 mL). Three mice were then placed in a 3-position bed holder (Hilton, Minerve) of the NanoScan^®^ PET/CT scanner (Mediso, Budapest, Hungary). Anesthesia was maintained using isoflurane (1.6–2.2% *v*/*v*) in a mixture of air and oxygen (1:1, at 1 L/min). Extraction of nose-cone air was set to 20 mL/min.

Temperature and breathing were actively monitored using breathing pads and rectal temperature probes (SA Instruments, Inc., Stony Brook, NY, USA). Respiration module and control/gating module (SA Instruments, Inc.) relayed information to the monitoring software (PC-SAM32, SA Instruments, Inc., Stony Brooke, NY, USA). Temperature control was achieved using a multi-station temperature control unit and air pump (Minerve, Esternay, France).

A TRAIL-R2 agonist (MEDI3039, 0.1–0.4 mg/kg, Medimmune, Gaithersburg, MD, USA) or vehicle control (PBS) was administered intravenously 24 h prior to imaging [38].

### 2.5. Dynamic PET/CT Imaging of Tumor Cell Death Using ^68^Ga-C2Am

COLO205 tumor-bearing mice (N = 6) underwent CT+PET scanning, which was performed consecutively in the same imaging session (total scan time 2.5 h), before and 24 h following treatment with MEDI3039. Helical CT data were acquired for anatomical reference and for attenuation correction using Nucline (V2.01, Mediso). Images were acquired with a semicircular scan, with 180 projections. The X-ray energy was set to 55 kVp with 600 milliseconds exposure and 1:4 image binning. Images were reconstructed using Butterworth filtering with an isotropic voxel size of 213 μm. PET images, with a nominal isotropic resolution of 0.6 mm, were reconstructed using a dynamic protocol (Tera-Tomo 3D (Mediso) algorithm), energy window 400–600 keV, coincidence modes 1–5, full detector model, normal regularization, 2 iterations, 6 subsets, with attenuation and random scatter correction. Images were analyzed in VivoQuant^®^ software (vs. 4.0 patch 1, InviCRO, Needham, MA, USA). A dynamic PET acquisition lasting 120 min was initiated 30 s prior to intravenous injection of 3.7 ± 1.2 MBq 68Ga-C2Am (1075 ± 382 μg protein/kg body weight; 10 mL/kg; SA = 0.15 ± 0.05 MBq/μg protein). Scans were reconstructed into 23 time bins (4 between 0–1 min; 4 between 1–5 min; 11 between 5–60 min and 4 between 60–120 min).

## 3. Results

### 3.1. Radiolabeling of C2Am Using Gallium-68 Chloride

A cysteine-reactive gallium-68 chelator (NODAGA-maleimide) was conjugated to C2Am using the single introduced cysteine residue, which is distant from the PS-binding site [25] (Figure 1). The NODAGA-labeled protein was loaded with gallium-68 using [^68^Ga]GaCl_3_ eluted from a generator, resulting in the quantitative production of ^68^Ga-C2Am.

### 3.2. Cell Binding Assays Using ^68^Ga-C2Am

Incubation of 68Ga-C2Am with MDA-MB-231 human breast cancer cells treated with a TRAIL R2 agonist (MEDI3039) [30,39] showed approximately four-fold greater retention of radioactivity than vehicle-treated control cells (Figure 2a) whereas MEDI3039-treated COLO205 human colorectal cancer cells showed approximately two-fold greater retention of activity than controls (Figure 2b). The activity retained by the cell pellets (Figure 2) was normalized to the total cell membrane surface area as the COLO205 cells formed small apoptotic bodies following MEDI3039 treatment [31]. As MEDI3039-treated COLO205 cells showed much less cell death-dependent contrast with 68Ga-C2Am than MDA-MB-231 cells, as was observed previously for ^18^F-C2Am [31], we elected to use these cells as a more challenging tumor model in which to evaluate ^68^Ga-C2Am as an imaging agent for detecting cell death in vivo.

### 3.3. Stability of ^68^Ga-C2Am

^68^Ga-C2Am was stable in PBS for up to 4.5 h (Appendix A) and in serum for up to 6 h at 37 °C (Appendix A), although there was some binding to serum albumin (Appendix A). Lower molecular weight metabolites (<3 kDa) appeared in the urine at 15 min (Appendix A), suggesting fast renal proteolysis and excretion of the imaging agent, as was also observed for ^18^F-C2Am [31].

### 3.4. Biodistribution of ^68^Ga-C2Am

Biodistribution was assessed in MEDI3039-treated tumor-bearing mice (Figure 3). Two hours following ^68^Ga-C2Am administration, most tissues showed low levels of activity (<3% ID/g), with the exception of the kidney, which is the main clearance route. Tumor activity post-treatment was ~50% greater than in every other organ, except the kidney, spleen, and liver. The relatively high levels of retention observed in the spleen and liver may have been due to drug-induced cell death, as was observed previously in studies with ^18^F-C2Am [31], and may be due to low-level expression of TRAIL-R2 in these tissues [40]. The blood half-life of 68Ga-C2Am, calculated from the analysis of dynamic data obtained from a region of interest located in the mouse carotid artery, was 25.0 ± 2.4 min (mean ± SEM, N = 5).

Regions of interest (ROI) were drawn manually for the delineation of tissue/organ ROIs. The fraction of injected dose per gram of tissue (ID/g, %), tumor-to-blood (T/b), and tumor-to-muscle (T/m) ratios were also calculated, the latter using lower flank muscle and the carotid artery, respectively.

### 3.5. PET-CT Imaging of Tumor Cell Death In Vivo Using ^68^Ga-C2Am

Dynamic PET/CT imaging (Figure 4) confirmed renal as the dominant excretion route for 68Ga-C2Am. Bladder signal could be detected within 15 min of administration (Figure 4a,b), and kidney cortical uptake increased up to 2 h post-injection (to ~25% ID/g; Figure 4c, green line). The tumor signal was detectable within 15 min of injection, and there was a small component (<3% ID/g) of hepatobiliary clearance (Figure 4c, purple line). T/m and T/b ratios were 2.2 ± 0.2 and 1.1 ± 0.1, respectively, at 2 h post-administration of 68Ga-C2Am. However, analysis of individual COLO205 tumors before and following treatment (Figure 4d) showed significant increases in tumor retention of 68Ga-C2Am at 2 h post-administration, expressed as (ID/g, %; *p* < 0.005), tumor-to-blood (T/b; *p* < 0.05), SUV or SUVmax (*p* < 0.005).

Standardized uptake values were calculated as maximum (SUVmax) and mean (SUV). SUV was defined Cimg/(ID/BW) where Cimg is the activity concentration (MBq/mL), ID is injected activity (MBq), and BW is body weight (g). SUVmax was calculated from the pixel with the maximum signal intensity.

### 3.6. Correlation of Tumor Contrast with the Levels of Cell Death Determined Histologically

Tumor contrast, expressed as the ratio of the signals post- and pre-treatment, was correlated with cleaved caspase 3 (CC3) staining (%) in tumor sections obtained following imaging. The contrast metrics ID/g (%), T/b, and SUV showed the best correlation with this histological marker of cell death (r > 0.6; Figure 5a). With increasing concentrations of MEDI3039 (0.1, 0.2, 0.4 mg/kg, 24 h, i.v.) there was an increase in the levels of tumor cell death (Figure 5b), and for all three metrics, a doubling of CC3 positivity correlated with an approximately 50% increase in the post/pre-treatment tumor contrast.

## 4. Discussion

Cell death imaging has been used to detect tumor treatment response and can provide prognostic information [5,6,7,8]. We have described ^68^Ga-labeled C2Am, which binds the PS exposed by apoptotic and necrotic cells, as a PET imaging agent for detecting tumor cell death following treatment and evaluate it here in comparison with the ^18^F- and ^111^In-labeled derivatives and with other PET agents for detecting cell death.

Following treatment of COLO205 tumors with a TRAIL-R2 agonist, the increase in tumor-to-muscle (T/m) contrast generated by 68Ga-C2Am was 2.3 ± 0.4, which was lower than that observed previously for 18F-C2Am (6.1 ± 2.1) [31], but identical to the contrast observed previously for the SPECT agent 111In-C2Am (2.2 ± 0.2) [29]. The mean tumor SUVmax following treatment (0.29 ± 0.04) was also lower than that reported for 18F-C2Am (0.40 ± 0.12) [31]. 68Ga-C2Am showed a longer blood half-life (25.0 ± 2.4 min) than 18F-C2Am (12.4 ± 2.2 min) [31] but similar clearance kinetics to 68Ga-duramycin (t1/2: 17.3 ± 4.12 min) [16]. The binding of 68Ga-C2Am to serum albumin may have contributed to the slower clearance of the agent when compared with the ^18^F-labeled C2Am derivative. Renal excretion was established as the main clearance route for 68Ga-C2Am, with minimal liver accumulation. As was observed for 18F-C2Am [31] we detected a small C2Am fragment (<3 kDa) in mouse urine at 15 min following administration, which is likely the result of proteolytic cleavage in the kidney and may contribute to the rapid excretion of the agent.

C2Am-based imaging agents can detect both apoptosis and necrosis and therefore should be more sensitive than those agents that detect apoptosis alone, such as ^18^F-ML-10 or ^18^F-ICMT-11. Moreover, C2Am is also capable of binding PE [41], which like PS, is also externalized to the surface of dying cells, which should further improve the sensitivity of this agent for detecting cell death. The levels of tumor cell death observed in the clinic can vary from a few percent pre-treatment with some breast cancer patients showing more than 10% apoptosis following neoadjuvant treatment [42] Therefore, we anticipate being able to detect this increase in apoptosis based on the findings shown in Figure 5, where a doubling of CC3 positivity resulted in a 50% increase in the post/pre-treatment tumor contrast.

## 5. Conclusions

We have demonstrated ^68^Ga-C2Am as a PET agent for imaging cell death in vivo. The agent showed a predominantly renal clearance and tumor contrast post-treatment, within 2 h of administration. Considering the wide availability of gallium-68 generators globally, ^68^Ga-C2Am has the potential to be used in the clinic to assess early response to cancer treatment.

## Figures and Tables

**Figure 1 cancers-15-01564-f001:**
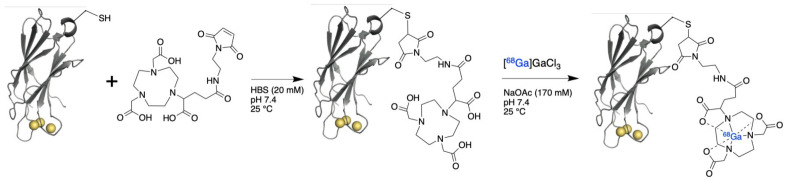
Radiochemical labeling of C2Am, using [^68^Ga]NODAGA-maleimide, to yield ^68^Ga-C2Am. The rapid Michael addition reaction was performed under mild conditions in HEPES-buffered saline (20 mM HEPES, 150 mM NaCl, pH 7.4) at 25 °C. The C2Am active site containing three calcium ions (yellow spheres) and the single cysteine residue (-SH) are indicated.

**Figure 2 cancers-15-01564-f002:**
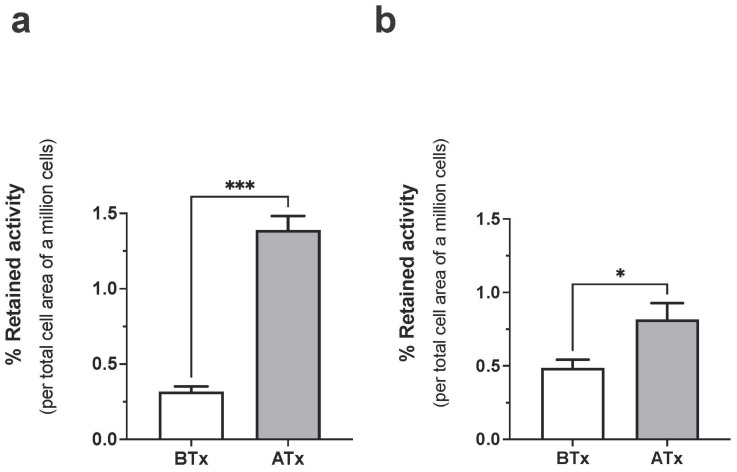
Binding of ^68^Ga-C2Am to untreated (BTx) and drug-treated (ATx) MDA-MB-231 (**a**) and COLO205 (**b**) cells. Fraction of retained activity (%), normalized to total membrane surface area is shown for both cell lines. Two-tailed, pairwise *t*-test, unequal variance. * *p* < 0.05, *** *p* < 0.0005, Retained activity (%) (mean ± SEM, N = 3 per group). BTx, Atx, before and after treatment, respectively.

**Figure 3 cancers-15-01564-f003:**
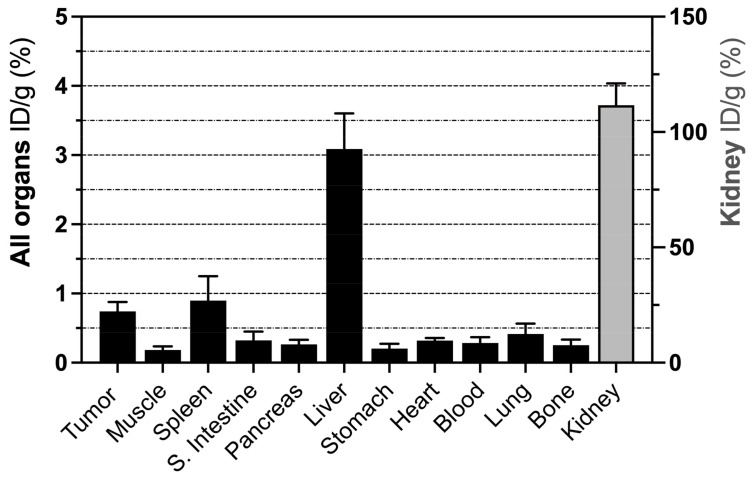
Biodistribution profile of ^68^Ga-C2Am. COLO205-tumor-bearing mice were treated with MEDI3039 (0.4 mg/kg, i.v., 24 h before ^68^Ga-C2Am administration). ^68^Ga-C2Am was injected (5 MBq, ~1 mg/kg, i.v.) and tissues collected post-mortem 2 h following probe administration. Fraction of injected dose per gram of tissue (ID/g, %) (mean ± SEM, N = 5).

**Figure 4 cancers-15-01564-f004:**
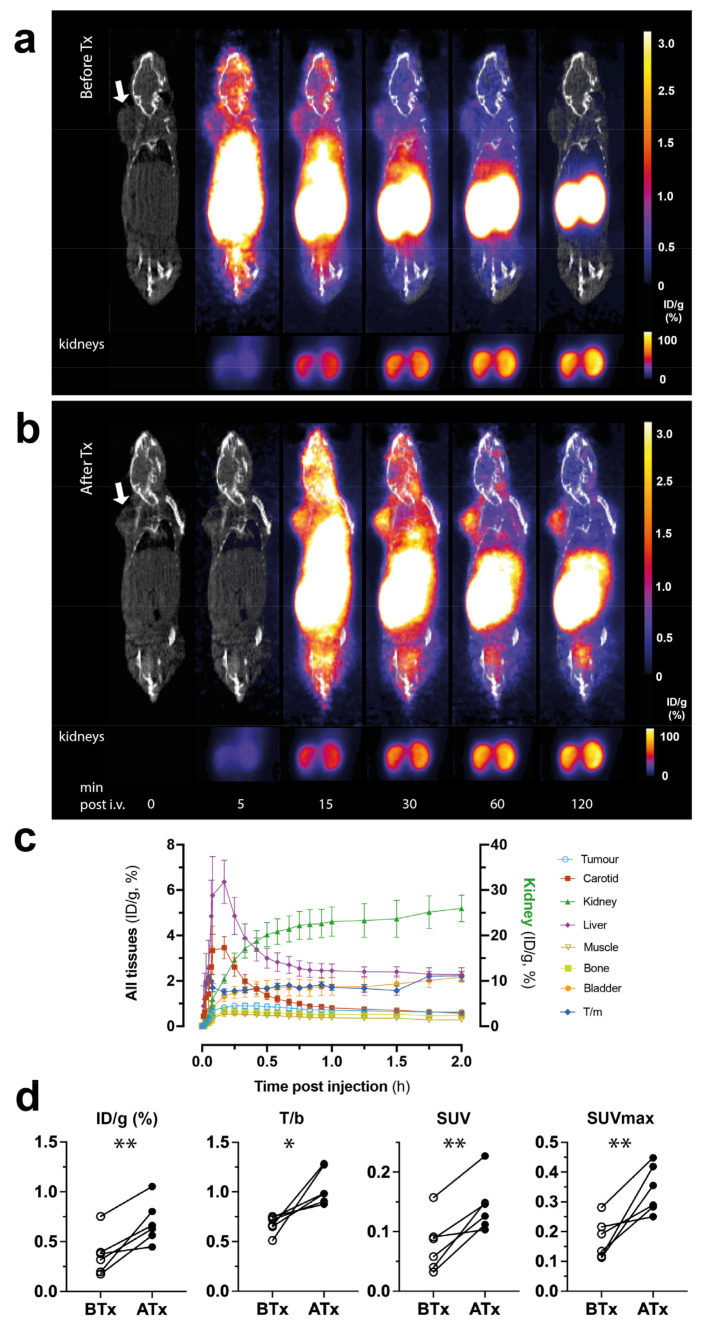
PET/CT images of ^68^Ga-C2Am accumulation in subcutaneous COLO205 tumors (white arrows). (**a**) Before treatment (BTx) and (**b**) 24 h after treatment with MEDI3039 at 0.4 mg/kg (Atx). Maximum intensity projections from representative mice. Images were acquired for up to 120 min after intravenous injection of ^68^Ga-C2Am. Signal (ID/g, %) is overlaid on a skeleton mask derived from CT images. (**c**) Activities (ID/g, %) in the indicated tissues in drug-treated animals (MEDI3039, 0.1–0.4 mg/kg, 24 h). (**d**) Pairwise analysis of PET signals before and 24 h after treatment at 2 h after probe administration, expressed as ID/g (%), tumor-to-blood (T/b) ratio, SUV, and SUVmax. Two-tailed, pairwise *t*-test, * *p* < 0.05, ** *p* < 0.01. Data in (**c**,**d**) are mean ± SEM, N = 6.

**Figure 5 cancers-15-01564-f005:**
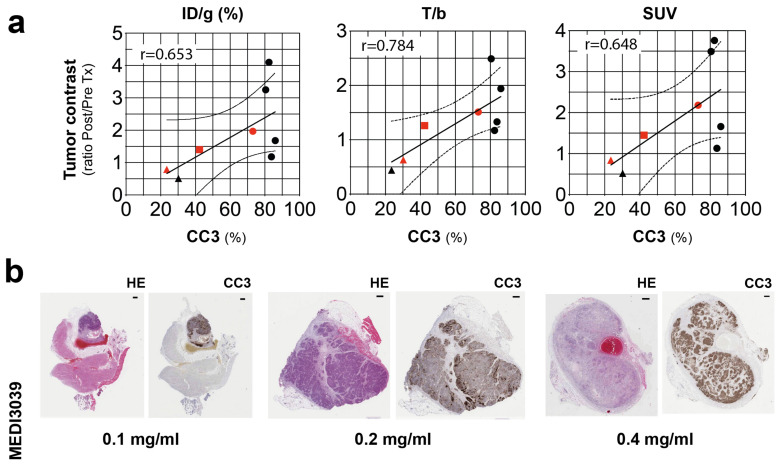
Correlation analysis of tumor PET signal with the histological marker of apoptosis, CC3 staining (%). (**a**) Pearson correlation values (r) are shown (top left corner of charts) for tumor contrast, expressed as ID/g, %, T/b, and SUV, 2 h post-administration of ^68^Ga-C2Am, vs. levels of tumor cell death (CC3, %). CC3 levels were estimated from two sections (200 μm apart) from MEDI3039-treated COLO205 tumors. Tumor contrast is expressed as the signal ratio, post/pre-treatment. COLO205 tumor-bearing mice (N = 8), treated with MEDI3039 (0.1—triangles, 0.2—squares, or 0.4 mg/kg, circles, 24 h, i.v.) (**b**) CC3 staining is shown for representative tumor sections corresponding to red symbols in (**a**). Two-tailed, r values are shown in (**a**). CC3, cleaved caspase 3; T/b, tumor-to-blood ratio.

## Data Availability

The data presented in this study are available on request from the corresponding author, details see reference [43].

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
