# Peer review of "Preclinical PET Imaging of Tumor Cell Death following Therapy Using Gallium-68-Labeled C2Am"

_cancers, 2023, doi:10.3390/cancers15051564_

Round 1
Reviewer 1 Report
This paper presents the radiotracer development for imaging tumor cell apoptosis after treatment. This imaging will be a valuable tool for the evaluation of various cancer treatments, such as chemotherapy and radiotherapy. In particular, It describes the development of [68Ga]C2Am radiotracer, which targets the phosphatidylserine-binding protein. The paper evaluates [68Ga]C2Am radiotracer, and its performance does not look good as [18F]C2Am radiotracer. However, its development is well-organized, so it meets the scope of this journal.
The reviewer also suggests the following minor errors.
1. Authors spelled "labeled" and "labelled" together. Please unify either way.
2. Line 55: 18F --> 18F
3. Line 79: Please define HBS.
4. ROI should be defined in Line 120 instead of Line 289.
5. SUV should be defined in Line 140 instead of Line 290.
6. %ID/g should be defined in Line 113 instead of Line 294.
7. Lines 177 and Line 180: Please check the reference format. [ ] --> ( )
8. Line 183: Please define PE.
9. Line 229: Please define specific activity. The authors use SA after this.
10. Line 258: Please check if this equation is correct.
11. Line 266: Please define PBS.
12. Line 283: 18 --> 68
13. Line 284: Is SA the correct expression?
14. Line 295: T/b T/m are defined previously.
Author Response
This paper presents the radiotracer development for imaging tumor cell apoptosis after treatment. This imaging will be a valuable tool for the evaluation of various cancer treatments, such as chemotherapy and radiotherapy. In particular, It describes the development of [68Ga]C2Am radiotracer, which targets the phosphatidylserine-binding protein. The paper evaluates [68Ga]C2Am radiotracer, and its performance does not look good as [18F]C2Am radiotracer. However, its development is well-organized, so it meets the scope of this journal.
The reviewer also suggests the following minor errors.
- Authors spelled "labeled" and "labelled" together. Please unify either way.
[Response] Corrected throughout the manuscript to ‘labeled’.
- Line 55: 18F --> 18F
[Response] Corrected
- Line 79: Please define HBS.
[Response] Defined as ‘HEPES-buffered saline (20 mM HEPES, 150 mM NaCl, pH 7.4)’
- ROI should be defined in Line 120 instead of Line 289.
[Response] ROI definition has been moved to line 352, immediately after the explanation of how blood half-life was calculated from carotid ROIs. The introduction has been modified and therefore lines differ from original manuscript.
- SUV should be defined in Line 140 instead of Line 290.
[Response] SUV definition moved to line 388 after description of SUV and SUVmax increase post treatment. The introduction has been modified and therefore lines differ from original manuscript.
- %ID/g should be defined in Line 113 instead of Line 294.
[Response]%ID/g definition moved to line 353 after the description of probe biodistribution. We did not use %ID/g in the cell uptake experiment hence it was not moved to line 113. The introduction has been modified and therefore lines differ from original manuscript.
- Lines 177 and Line 180: Please check the reference format. [ ] --> ( )
[Response] Journal guidelines suggest using [ ]. All refs are now formatted now to [ ]
- Line 183: Please define PE.
[Response] Line 455 defined as ‘phosphatidylethanolamine’
- Line 229: Please define specific activity. The authors use SA after this.
[Response] Line 713 defined.
- Line 258: Please check if this equation is correct.
[Response] Line 744 checked and is correct. Surface area of sphere is 4 ? r 2 . If we use the diameter instead of radius then formula is 4 ? (D/2 ) 2 -> 4 ? D 2/4 as per manuscript or to simply further ? D 2
- Line 266: Please define PBS.
[Response] Line 751 defined as ‘phosphate-buffered saline (0.1 mL, 137 mM NaCl, 2.7 mM KCl, 8 mM Na2HPO4, and 2 mM KH2PO4)’
- Line 283: 18 --> 68
[Response] Line 786 corrected to 68Ga from 18Ga.
- Line 284: Is SAthe correct expression?
[Response] Line 804 corrected to ‘SA’.
- Line 295: T/b T/m are defined previously.
[Response] Line 353 T/b and T/m are now defined. All other definitions (e.g. line 169) have been removed.
Reviewer 2 Report
The article present a good interest but it must be clearly reworked. You must review the whole article. There are errors and important informations are missing.
Introduction is too simple without real introduction about the article. We would like to explanation about the target. Interest of this radiotracer need to be more developed. Too long sentences. PET imaging resolution is not allow to see cell, we are around 2 mm for clinical PET, it represent at least 200 cells. FDG is used above 90% in clinical routine, FDG is interesting because is an analogue of glucose and cancer cells are avid of glucose. The authors write: “has been inhibited by the…” line 58 to 60. Fluor 18 radionuclide present a real advantage with possibility to have high activity production, that is very interesting for clinic to inject several patients with one production. Lot of radiotracer labeled with fluor 18 are also developed.
Result: no results on radiolabeling are presented RCP, AY, pH, acticity of final product... The other results are quite well presented
Materials and Methods: Manually radiolabeling? No explications about quality control of radiolabeling, no TLC? Are you shure to see free gallium, impurity with your HPLC method? How many mice for the study, n=5 for dynamics, 6 point in figure 4d ?
Discussion: needing to reviex with in put literature, annexin V has been labelled with 18F and 68Ga...
Line 60: gallium 68 half-life = 67,8min
Line 155: "capase" caspase
Figure 2: no asterisk?
Figure 4 d "Pairwise analysis of PET signals in tumor before and 24 h ...
Different titles between article and supplementary data
supplementary data: table below graph is to small removed and make one readable, only with the useful data
Author Response
The article present a good interest but it must be clearly reworked. You must review the whole article. There are errors and important informations are missing.
Introduction is too simple without real introduction about the article. We would like to explanation about the target. Interest of this radiotracer need to be more developed. Too long sentences. PET imaging resolution is not allow to see cell, we are around 2 mm for clinical PET, it represent at least 200 cells. FDG is used above 90% in clinical routine, FDG is interesting because is an analogue of glucose and cancer cells are avid of glucose. The authors write: “has been inhibited by the…” line 58 to 60. Fluor 18 radionuclide present a real advantage with possibility to have high activity production, that is very interesting for clinic to inject several patients with one production. Lot of radiotracer labeled with fluor 18 are also developed.
[Response] Lines 52-68 We have added remarks about the PS target and its concentration and removed the discussion of FDG-PET, which was not entirely relevant in the context of this paper. We are not aiming to observe individual cells but rather whole tumors responding to therapy therefore the clinical image resolution of PET is not a problem. Fluorine-18 has indeed its own advantages however it comes at a high cost and for labeling C2A requires a cyclotron to be nearby. Radiometals can be eluted from a generator are a cheaper and faster alternative to 18F. There is a full discussion of other cell death imaging agents in the discussion section of the manuscript.
Result: no results on radiolabeling are presented RCP, AY, pH, activity of final product... The other results are quite well presented
[Response] Radiochemical purity (RCP) is shown in Supporting Figure 1 (top).
Line 714: The pH was adjusted using NaOH (0.1M) to a final pH between 7 and 7.5.
Line 715: Activity yield (AY) at the end-of-synthesis was high >95% due to short incubation time.
Line 716: Activities of final product at the EOS ranged between 28-34 MBq and was highly dependent on the activity of generator eluate.
Materials and Methods: Manually radiolabeling? No explications about quality control of radiolabeling, no TLC? Are you shure to see free gallium, impurity with your HPLC method? How many mice for the study, n=5 for dynamics, 6 point in figure 4d ?
[Response] Line 499 & 714 Low levels of activity were used therefore no automation was needed/nor available for this study therefore chelation of 68Ga was performed manually in a lead castle.
[Response] Line 720: Our research group uses UPLC+radiodetectors for QC. The column used (Superdex 75 Increase 5/150 gel filtration) is used for separating low molecular weight proteins (3 kDa to 70 kDa). Small molecules (including [68Ga]GaCl3) have a high retention time on the column. This provides the best separation between the labeled 68Ga-C2am and free GaCl3. We are confident, therefore that the levels of free gallium are minimal. The following text has been added to the revised manuscript at lines 718: Quality control consisted of Ultra Performance Liquid Chromatography using a Superdex 75 Increase 5/150 gel filtration column, which is designed for separating low molecular weight proteins (3 kDa to 70 kDa). Small molecules, including [68Ga]GaCl3, have long retention times (>10 minutes).
[Response] Line 408: Six mice were used. This information has been added to the revised manuscript.
Discussion: needing to reviex with in put literature, annexin V has been labelled with 18F and 68Ga...
[Response] We have already shown that 18F-C2am outperforms 68Ga-C2Am in terms of tumour contrast by about 3-fold. Therefore, we do not expect 68Ga-C2Am to outperform other 18F labeled radiotracers (e.g. Annexin). We have discussed other cell death imaging agents in the discussion section, including Annexin. The following text has been added to the revised manuscript.
Line 462: Two 68Ga-labeled derivatives of Annexin V were produced by site-specific labeling at introduced cysteine residues. The derivatives showed rapid, predominantly renal clearance, with short blood half-lives of ~6 minutes, however kidney retention was high [20]. An 18F-labeled Annexin V derivative, labeled specifically at an introduced cysteine residue, showed much lower renal retention [21].
Line 60: gallium 68 half-life = 67,8min
[Response] Line 64: Corrected to 67.8 min
Line 155: "capase" caspase
[Response] Line 412: corrected
Figure 2: no asterisk?
[Response] Line 320: ***P<0.0005 added
Figure 4 d "Pairwise analysis of PET signals in tumor before and 24 h ...
[Response] Line 405: Corrected to Pairwise analysis of PET signals before and 24 h after treatment ‘at’ 2 h after probe administration
Different titles between article and supplementary data
[Response] corrected
supplementary data: table below graph is to small removed and make one readable, only with the useful data
[Response] Separate tables with useful data have been generated where appropriate.